# A Sequencing Overview of Malignant Peripheral Nerve Sheath Tumors: Findings and Implications for Treatment

**DOI:** 10.3390/cancers17020180

**Published:** 2025-01-08

**Authors:** Kangwen Xiao, Kuangying Yang, Angela C. Hirbe

**Affiliations:** Division of Oncology, Department of Internal Medicine, Siteman Cancer Center, Washington University School of Medicine, St. Louis, MO 63110, USA; kangwen@wustl.edu (K.X.); kuangyingyang@wustl.edu (K.Y.)

**Keywords:** MPNST, genomics, RNA-seq, targeted therapy, epigenetics

## Abstract

In recent decades, advancements in high-throughput sequencing technology, coupled with reduced sequencing costs, have led to a significant increase in the genomic profiling of benign and malignant peripheral nerve sheath tumors. This review synthesizes recent sequencing discoveries from multiple sequencing technologies, underscores the critical mutation events involved in tumor pathogenesis, and explores their potential therapeutic implications. By elucidating the molecular basis of these tumors, clinicians and researchers can improve patient outcomes and provide a foundation for more effective treatment strategies.

## 1. Overview of MPNSTs

The earliest MPNST in a human can be traced back to 1952, which was described as an unclassified tumor which resembles a neurofibroma in a Caucasian male [1]. In 1984, Ducatman et al. reported 16 cases of MPNSTs in children under 16, with a mean survival of only 1.8 years [2]. In 1986, a study reviewed 120 cases of MPNSTs and concluded that total resection of the tumor could improve the prognosis of the MPNST. Surgery is still the main treatment for MPNSTs today, but recurrence rates are high [3]. MPNSTs predominant locate along the main nerve bundles [4]. While MPNSTs are traditionally characterized as Schwann cell-derived tumors, this may oversimplify their heterogeneous nature. A recent study highlights the fact that low-grade MPNSTs should be identified as ANNUBP with increased proliferation [5]. The presenting symptom of MPNSTs is typically a gradually enlarging painless mass, which, as the tumor grows to a certain size, compresses adjacent tissues and becomes painful [6]. The MPNST is ranked as the sixth most common soft tissue sarcoma, consisting of 2%~10% of soft tissue sarcomas, annually [7].

The treatment for MPNSTs depends on the stage of the disease. For localized MPNSTs, complete surgical resection with or without chemotherapy and radiation remains the most effective therapy to date [8,9]. A recent retrospective analysis of an MPNST cohort showed that some patients receiving radiation therapy had a better overall survival rate, indicating that radiation therapy may be beneficial for MPNSTs [10]. However, the effect on overall survival with MPNST requires further clinical validation [11]. For those patients who have already developed metastasis, chemotherapy—typically doxorubicin and ifosfamide-based regimen—remain the main treatment, but only extends life expectancy by 1 to 2 years [12,13]. The overall prognosis remains poor, despite surgical resection, adjuvant radiation, or chemotherapy, with the five-year overall survival rate of 47.2% [14].

NF1 is one of the most common tumor predisposition syndromes and is caused by the loss of function of the *NF1* tumor suppressor gene. Patients have an increased risk of developing benign tumors, as well as malignancies including soft tissue sarcoma, glioma, breast cancer, and melanoma [15]. Fifty percent of MPNSTs occur in patients with NF1, while 40% occur sporadically, and 10% occur in the setting of prior radiation [16]. The etiology of sporadic MPNSTs is not entirely understood, but the tumors are thought to arise de novo instead of from PNs and ANNUBP [17]. The pathogenesis of NF1-associated MPNST involves dysregulation of multiple genes and progression through several disease stages. In addition to the *NF1* gene loss, several other key genes including *CDKN2A*, *TP53* and *EED/SUZ12* have been identified as being involved in the pathogenesis of MPNSTs [18,19,20]. In the context of NF1, the loss of the second copy of the *NF1* gene along with cues from the heterozygous microenvironment leads to the development of a PN, which is a benign tumor that can cause pain, mobility dysfunction, and organ compromise, depending on the location [21]. The ANNUBP was proposed in 2017 and is an intermediate pre-malignant disease stage between PNs and MPNST [22]. It has been shown that loss of *CDKN2A* is correlated with ANNUBP [23] and further loss of PRC2 components, *TP53*, or other alterations are likely to drive the malignant transformation from ANNUBP to MPNST [24,25,26].

In the last several decades, with the rapid development of high-throughput sequencing technology and the reduction of sequencing costs, many MPNST samples have been sent for sequencing, including micro-array, RNA-sequencing (RNA-seq), methylation-sequencing, single-cell RNA sequencing, whole exome sequencing (WES) and whole genome sequencing (WGS). The complex genomic characteristics of MPNSTs are gradually being uncovered at multiple levels. Here we summarize recent MPNST-associated sequencing studies, highlighting potential targeted therapy options.

## 2. Sequencing Technologies and Its Application in MPNST

Researchers have utilized sequencing studies on MPNSTs with several research focuses. (1) Identification of molecular subtypes and diagnostic tools of MPNST. (2) Comparison of differential gene expression between MPNST and peripheral nerve sheath tumor (PNST). (3) Investigation of the pharmacogenomic changes following various drug treatments. (4) Identification of functional genes in MPNSTs by analyzing genetic mutations and gene-noncoding RNA interactions. The following chapters will introduce in detail the application of various sequencing technologies in MPNSTs. The summary of key findings is shown in Table 1 and Figure 1.

### 2.1. Microarray and RNA-Seq Analysis

Microarrays are based on hybridization of pre-designed labeled probes with target cDNA sequences [27]. To date, microarray sequencing technology is still one of the most common technologies due to its relatively low cost and straightforward pipeline and has been widely applied in MPNST samples. These datasets employ a range of platforms, including Affymetrix, Agilent, and Illumina. However, the accuracy of this technology relies heavily on the pre-designed probes, and the affinity of the probe hybridization [28]. Therefore, microarray technology is not the best fit for samples with low-abundance transcripts, and cannot distinguish isoforms or identify genetic variations. In addition, probes are often accompanied by problems such as cross-hybridization and non-specific hybridization [28].

**Table 1 cancers-17-00180-t001:** Recent sequencing summary of MPNST-related samples.

Data Type	Organism		Sample Settings				Reference
			MPNST	PN	NF	ANNUBP	
Microarray	Human	Tissue	64	NA	15	NA	GSE241224 (Høland et al., 2023) [29]
Microarray	Human	Tissue	6	10	9	10	GSE239561 (Rhodes, 2023) [30]
Microarray	Human	Tissue	10	NA	NA	NA	GSE52390, GSE52391 (Wolf et al., 2013) [31]
DNA methylation	Human	Tissue	10	NA	NA	NA	
Microarray	Human	Tissue	6	NA	26	NA	GSE41747 (Jessen et al., 2012) [32]
Microarray	Mouse	Tissue	18	NA	15	NA	
Microarray	Human	Tissue	16	NA	NA	NA	GSE17118 (Lafferty-Whyte et al., 2010) [33]
Microarray	Human	Tissue	3	NA	3	NA	GSE52252 (Wang et al., 2014) [34]
Microarray	Human	Tissue	30	NA	8	NA	GSE66743 (Kolberg et al., 2015) [35]
Microarray	Human	Tissue	4	NA	NA	NA	GSE77203 (Yasuhiro et al., 2019) [36]
Microarray	Human	Cell	21	NA	NA	NA	GSE39764 (Sun et al., 2013) [37]
Microarray	Human	Tissue	3	NA	NA	NA	GSE35852, GSE35851 (Kelly et al., 2012) [38]
micro-RNA	Human	Tissue	3	NA	NA	NA	
Microarray	Human	Cell	22	NA	NA	NA	GSE8717 (Mahller et al., 2007) [39]
Microarray	Human	Cell	20	NA	NA	NA	GSE47476 and GSE47477 (Zhang et al., 2013) [40]
micro-RNA	Human	Cell	12	NA	NA	NA	
Microarray	Human	Cell	9	NA	NA	NA	GSE62500 (De Raedt et al., 2014) [41]
Microarray	Human	Cell	12	NA	NA	NA	GSE84205 (Malone et al., 2017) [42]
Microarray	Human	PDX/Tissue	11	NA	NA	NA	GSE60082 (Castellsagué et al., 2015) [43]
Microarray	Mouse	Tissue	8	NA	NA	NA	GSE57141 (Malone et al., 2014) [44]
Bulk RNA-seq	Human	PDX	13	NA	NA	NA	syn11638893 (Hirbe et al., 2022) [45]
WES/WGS	Human	PDX	16	NA	NA	NA	
scRNA-seq	Human	PDX	7 MPNST				
Bulk RNA-seq	Human	Tissue	73	NA	1	2	EGAD00001008608 (Genomics of Malignant Peripheral Nerve Sheath Tumor (GeM) Consortium, 2020) [46]
WGS	Human	Tissue	72	NA	2	3	
Bulk RNA-seq	Human	Tissue	41	NA	NA	NA	GSE206527 and GSE179699 (Chi et al., 2022) [47]
ATAC-seq	Human	Cell	12	NA	NA	NA	
Bulk RNA-seq	Human	Tissue	9	NA	8	NA	GSE178989, GSE179033 and GSE179041 (Wu et al., 2022) [48]
scRNA-seq	Human	Tissue	PN vs. MPNST				
scRNA-seq	Mouse	Tissue	1.5-month MPNST vs. 4-months MPNST				
DNA methylation	Human	Cell	63	NA	NA	NA	GSE141438, GSE141435 and GSE141437 (Kochat et al., 2021) [49]
Bulk RNA-seq	Human	Tissue	7	NA	3	NA	
Bulk RNA-seq	Human	Cell	36	NA	NA	NA	
Bulk RNA-seq	Human	Tissue	25	21	NA	NA	GSE145064 (Kohlmeyer et al., 2020) [50]
Bulk RNA-seq	Human	Tissue	12	12	NA	NA	GSE212964 (Vasudevan et al., 2023) [51]
scRNA-seq	Human	Tissue	3 MPNST vs. 3 PN				
Bulk RNA-seq	Human	Tissue	14	NA	34	NA	GSE207400, PRJNA854920 and GSe207399 (Suppiah et al., 2023) [52]
WES	Human	Tissue	18	NA	16	NA	
scRNA-seq	Human	Tissue	ANNUBP vs. MPNST				
Bulk RNA-seq	Human	Tissue	10	NA	NA	NA	TCGA-SARC (https://www.cancer.gov/tcga, accesed on 28 August 2024)
Bulk RNA-seq	Human	Tissue	6	NA	NA	NA	GSE120685 (Woodhoo et al., 2021) [53]
Bulk RNA-seq	Human	Cell	28	NA	NA	NA	GSE183308, GSE183307 (Zhang et al., 2022) [54]
scRNA-seq	Human	Tissue	primary vs. metastasis				
Bulk RNA-seq	Human	Tissue	3	NA	3	NA	GSE270880 (Zhang et al., 2024) [55]
Bulk RNA-seq	Human	Cell	16	NA	NA	NA	GSE179585, GSe179586 (Patel et al., 2022) [56]
DNA methylation	Human	Cell	16	NA	NA	NA	
Bulk RNA-seq	Human	Cell	6	NA	10	NA	GSE118185 (Wassef et al., 2019) [57]
Bulk RNA-seq	Human	Cell	6	NA	NA	NA	GSE213988 (Chung et al., 2022) [58]
Bulk RNA-seq	Human	Cell	4	NA	NA	NA	GSE216792
WGS	Human	Tissue (blood)	46	23	NA	NA	syn23651229 (Shern et al., 2020) [59]
WES	Human	Tissue	51	NA	NA	NA	EGAS0000100452 (Lyskjær et al., 2020) [60]
WES	Human	Tissue	2	3	NA	NA	Hirbe et al., 2015 [61]
WES	Human	Tissue	15	NA	NA	NA	Lee et al., 2014 [19]
WES	Human	Tissue	6	NA	NA	NA	Godec et al., 2020 [62]
WES/WGS	Human	Tissue	11	NA	2	NA	Kinoshita et al., 2020 [63]
DNA methylation	Human	Tissue	1	NA	1	NA	GSE21714 (Feber et al., 2011) [64]
DNA methylation	Human	Cell	16	NA	NA	NA	GSE263127 (Bhunia et al., 2024) [65]
DNA methylation	Human	Tissue	102	NA	NA	NA	E-MTAB-8864 www.ebi.ac.uk/biostudies/arrayexpress/studies/E-MTAB-6961, accesed on 12 December 2024)
DNA methylation	Human	Tissue	8	NA	NA	NA	GSE36982 (Renner et al., 2012) [66]
micro-RNA	Human	Tissue	19	9	NA	NA	GSE140987 (Wiemer, 2019) [67]
ATAC-seq	Human	Cell	6	NA	NA	NA	GSE275047

NA: Not applicable; PDX: Patient-derived xenograft; ATAC-seq: Assay for Transposase-Accessible Chromatin with sequencing.

Given the limitation of microarray sequencing, RNA-seq, one of the most common next-generation sequencing technologies, has been widely applied in the field of biomedicine [68]. RNA-seq has several advantages: first, it can detect low-abundance transcripts. Second, it does not rely on designed probes or primers in advance. Thirdly, new transcripts and splice variants can also be discovered [69].

Molecular classification has become increasingly important in tumor diagnostics and treatment. For instance, in 2016, medulloblastoma was subdivided into multiple molecular subtypes, including the Sonic Hedgehog (SHH) and WNT subtypes, based on transcriptomic data [70]. This subtype-specific classification has since enabled the development of tailored therapeutic strategies, significantly improving patient prognoses [71]. Similarly, several studies have utilized computational algorithms to identify different gene expression patterns to define subtypes of MPNST. For example, Holand et al. utilized the transcriptome of the MPNST to perform clustering analysis and identified two immune-related clusters of MPNSTs which correlated with patient prognosis [29]. They predicted that *EGFR*, *EZH2*, *KIF11*, *PLK1* and *RRM2* could be potential therapeutic targets for MPNSTs. The largest MPNST sequencing study to date was performed by the GeM consortium, which suggested that genomic expression pattern could predict the prognosis of MPNST better than clinical or pathological evidence alone [72]. In 2023, Suppiah et al. identified two distinct MPNST subtypes, which were associated with activated SHH pathway and WNT pathway, respectively. They found that further targeting of the SHH pathway inhibited growth of MPNST with poor prognosis [52]. Taken together, these analyses highlight the heterogeneity of MPNSTs and revealed different gene expression signatures, which may inform the precision medicine of MPNSTs in the future.

Other studies have focused on comparison between MPNSTs and normal adjacent samples, PN, ANNUBP and benign NF, respectively. For example, Kohlmeyer et al. identified the upregulation of genes such as *TWIST1*, *AURKA*, *BUB1*, and *NEK2* in MPNSTs compared to PN, which appears to be important for MPNST growth and proliferation [50]. The comparison of MPNSTs with ANNUBP by Mitchell et al. identified *CCMB1*, *WNT3A*, *GAS1*, and *PROM1* as significantly upregulated genes [30]. It is plausible that these could be key genes during the transformation of ANNUBP to MPNST. Wu et al. noted the upregulation of epithelial–mesenchymal transition (EMT) and *EZH2* pathways in MPNSTs and upregulation of P53 and interferon signaling in benign neurofibromas (NF) [48]. However, few recurrent DEGs or pathways are identified among these studies, highlighting the heterogeneity throughout MPNST development.

Biomarkers related to Schwann cell development have been identified as potential predictive and diagnostic tools for MPNST. Recently, Holand et al. found *CDH19*, *ERBB3*, *S100B* and other Schwann cell differentiation markers were downregulated in MPNSTs compared to neurofibroma, indicating dysregulation of Schwann cell development in MPNSTs [29]. Vasudevan et al. added to this by showing that compared to neurofibroma cells, Schwann cell differentiation markers *S100B* and *SOX10* were downregulated in PRC2 mutant MPNST cells, suggesting that PRC2 loss is related to the transformation of Schwann cells into MPNSTs [51]. The above sequencing data have shown distinct gene expression signatures between other PNST and MPNSTs and identified a few gene candidates for MPNSTs. However, there is still a lack of RNA-seq-based predictors to clinically assess the risk of patients transforming from PNs to MPNST and then formulate personalized treatment plans. In addition, there is also a lack of a consistent set of genes or pathways implicated across all studies, which highlights the inherent complexity and heterogeneity of MPNSTs. Future research needs to focus on identifying common molecular pathways that can be targeted for therapy, as well as developing reliable predictive models based on RNA-seq data in a prospective manner to guide clinical decisions.

Gene expression analysis can also help assess the drug synergy effect and uncover potential mechanisms to target the MPNST. For example, Borcherding et al. treated MPNST cells with *TYK2* inhibitor, and then performed RNA-seq which showed the upregulation of the MAPK pathway after treatment. Following this finding, they combined mirdametinib, a MEK inhibitor, with deucravacitinib, a *TYK2* inhibitor, and these two drugs synergistically inhibited the MPNST [73]. In another example, Sun et al. found that the BMP2-SMAD1/5/8 pathway was activated in NF1-associated MPNSTs, and inhibition of this pathway reduced tumor cell proliferation and invasion, hinting at a possible therapeutic target [37]. These studies have uncovered novel therapeutic targets for MPNSTs, and guide more effective treatment strategies.

Overall, microarray and RNA-seq have greatly deepened our understanding of the pathogenesis of MPNSTs at the gene expression level. However, traditional bulk RNA-seq extracts mixed RNA from tissues or a group of cells for sequencing to obtain the average gene expression of all cells, which cannot reveal the expression specificity of individual cells, and cannot understand the interaction between tumor cells and other cells in their microenvironment and the role of each cell type. Future studies should be focused on combination with single-cell and spatial transcriptomic approaches to dissect the cellular heterogeneity within the MPNST.

### 2.2. Single-Cell RNA-Seq

The single-cell RNA-seq reveals the gene structure and gene expression level of individual cells in the MPNST, reflecting the heterogeneity of cells and allowing for analysis of the contribution of individual cells to the MPNST. Several studies have revealed that MPNSTs are composed predominantly of tumor cells, whereas PNs show a higher proportion of immune-rich environments. For example, Vasudevan et al. showed that MPNSTs were mainly enriched in tumor proliferation cell clusters and growth factor-stimulated tumor cell clusters, while PNs were mainly enriched in T cell and endothelial clusters, which are non-tumor cells [51]. Zhang et al. showed a higher proportion of MPNST cells and fewer immune cell clusters in metastatic MPNSTs, compared to primary MPNSTs, which suggests that metastatic MPNSTs may suppress the immune environment more effectively than even the primary tumors, leading to their aggressiveness and ability to evade immune surveillance [54].

Similar trends are observed in mice models. Wu et al. found that the microenvironment in early-tumor stages of MPNST had more antitumor characteristics, with a higher presence of antitumor macrophages, while the microenvironment in advanced tumor stage became more pro-tumorigenic, characterized by a rise in activated fibroblasts and anti-inflammatory macrophages. They also compared the PNs with MPNSTs and determined unique cell clusters including hypoxic malignant Schwann cell precursors, mesenchymal-like neural crest cells, and mesenchymal-like malignant cells in MPNST, while immature Schwann cells as well as mesenchymal-associated macrophages were enriched in PNs [48].

In another study, Suppiah et al. found that MPNSTs with worse prognosis had a larger proportion of neoplastic cells, while in MPNSTs with better prognosis and atypical NF, the proportion of immune cells are significantly higher. They also found that the known marker of Schwann cells including *S100B* and *ERBB3* were decreased in MPNSTs with worse prognosis, while early neural crest cell markers *SOX9* and *OTX2* were increased in MPNSTs with better prognosis, indicating that the de-differentiation of Schwann cells is related to the progression to MPNSTs [52].

The above analysis showed that unique cell types, including malignant Schwann cell precursors and mesenchymal-like cells, have been identified in MPNSTs, while Schwann cells and macrophages dominate in PNs. These findings highlight the cellular heterogeneity of MPNSTs and their evolving tumor microenvironment as the disease progresses. However, several limitations of single-cell sequencing technologies cannot be ignored: first, the low-expression transcripts are still likely to escape the detection from sequencing. Second, it requires good practice and relatively complex training to obtain a qualified dataset. Third, it is easy to produce artifacts such as multiplets. Finally, it mainly focuses on the expression of different types of cells and ignores the spatial information and physical interaction of different cells in the tumor microenvironment. Future research should strive to improve the sensitivity of the low-abundance transcripts and simplify the pipeline of single-cell RNA-seq.

### 2.3. Whole Exome Sequencing (WES) and Whole Genome Sequencing (WGS)

WES and WGS have been widely used to uncover MPNST mutation patterns. NF1 germline mutation and the inactivation of this tumor suppressor gene, is known to predispose individuals to various cancers, including MPNSTs. Additional somatic mutations in NF1 can lead to biallelic inactivation, further increasing the malignancy rate of the disease [74]. In 2015, Hirbe et al. sequenced the same MPNST patient with different stages of diseases including PN, MPNST and metastasis MPNST. They found that the proportion of NF1 mutations increased with the progression of the MPNST and that the tumors became much more copy-number aberrant as they progressed to cancer [61].

PRC2 complex mutations, *TP53* loss, and *CDKN2A* loss are other common events in MPNSTs. In 2021, Dehner et al. performed whole exome sequencing in MPNST PDX lines. They found that *SUZ12* loss occurred in 62.5% of PDX lines while proportion of *TP53* loss was only 12.5% [45]. Similarly, Lee et al. noticed that the proportion of PRC2 components *EED/SUZ12* loss in sporadic MPNST, NF1-MPNST, and radiation-induced MPNST was 92%, 70%, and 90%, respectively. In addition, they found that somatic mutation of *CDKN2A* co-occurred with PRC2 loss, accounting for 81% of all MPNSTs [19]. Cortes-Ciriano et al. also showed that histone H3 on lysine 27 (H3K27me3) loss occurred in 55% of MPNSTs and biallelic inactivation of *SUZ12* and *EED* occurred in 28% and 17% of MPNSTs, respectively. They also found that *CDKN2A* biallelic inactivation occurred in 63% of NF1-related MPNST and 55% of sporadic MPNST [72]. In Lorenz’s study, CDKN2A loss existed in 100% of MPNSTs (n = 15) and 100% of MPNST cell lines (n = 8) [75]. These recurrent gene mutation events also suggest that these genes are of great significance in the development of MPNSTs.

Chromosome aberrations have been found in MPNSTs, especially the chromosome 8 gain. Szymanski et al. showed extensive chromosome aberrations, including gain of 1q, 7p, 8q, 9q, and 17q and chromosome loss in 6p and 9p in MPNST [76]. Similarly, Suppiah et al. also showed that gain of *SMO*, 1q, 8q,13q, 17p and 18q occurred in MPNSTs with poor prognosis [52]. Dehner et al. also found that chromosome 8 gain was exclusively occurring in 87.5% of MPNSTs, while no chromosome 8 gain was observed in PN samples, indicating chromosome 8 gain may be an important genomic event in MPNST development [45].

Together, the above sequencing studies displayed multiple events that occur during the transformation from PNs to MPNSTs, including loss of *NF1*, *CDKN2A*, PRC2 components and *TP53*, which are considered as common genetic mutation events in MPNSTs. Many other gene structural variations are identified such as chromosome gain of 8q, 1q, and 19q. However, it remains unclear at which stage these structural variations occur. Additionally, to understand polyclone evolution during the MPNST transformation, it would require more spatial transcriptome and genome sequencing evidence.

### 2.4. Epigenetics Sequencing

Epigenetic alterations, including DNA methylation, histone modifications, and non-coding RNA regulation, play a critical role in the pathogenesis of MPNSTs, as well.

Few recurrent gene methylations are reported in MPNSTs. *CDKN2A*, however, has been identified as one of the few recurrent hypermethylated genes in MPNSTs. In 2011, Feber et al. found hypermethylation of *CDKN2A* in MPNSTs, which indicated that the repression of the *CDKN2A* might be due to the epigenetic dysregulation in some cases [64]. In addition, Renner et al. found that CpG sites of *CDKN2A* could reflect the different types of soft tissue sarcomas [66]. Other distinct DNA methylations in MPNSTs have been also identified. For example, several differentially methylated CpGs located in *IL17* were shown to be related to MPNST progression [77]. Danielsen et al. detected hypermethylation of promoter of *RASSF1A* in 60% of MPNST samples, which is also correlated with poor prognosis of NF1-associated MPNST, indicating *RASSF1A* might be a prognostic predictor in a subset of MPNSTs [78]. These studies have implications for the epigenetic-based therapies of MPNSTs.

In addition to CDKN2A alterations, mutations in components of the PRC2 complex are common in MPNSTs. PRC2 primarily catalyzes the methylation of H3K27me3 and has been identified as an important modification in MPNSTs. For example, Kochat et al. found that PRC2 loss would reduce the repression of several enhancers, which further promotes MPNST progression [49]. In 2016, Röhrich et al. found that loss or reduction of H3K27me3 was exclusively seen in a subset of MPNSTs based on methylation clustering, which could be used as a biomarker to differentiate the cellular schwannoma and MPNST [79]. Cortes-Ciriano et al. found that global hypomethylation occurred in MPNSTs and hypermethylation of CpG islands and PRC2-component mutations occurred in MPNSTs with poor prognosis compared to MPNSTs with a better prognosis in patients with NF1 [72]. In 2022, Yan et al. performed assay for transposase-accessible chromatin (ATAC-seq) to detect the chromosome accessibility after the knockout of *SUZ12*. The results showed that the loss of PRC2 due to the knockout of *SUZ12* led to decreased chromosome accessibility, which further impaired IFN-gamma response in MPNST cells [47]. However, a previous study showed that was no significant survival difference between MPNSTs with the loss and retaining of PRC2 (n = 100), but they did not stratify patients by NF1 status, which may account for the lack of correlation with survival [60].

Micro-RNA array sequencing has been applied in MPNSTs recently. Amirnasr et al. identified upregulated micro-RNAs miR135b and miR-889 in MPNSTs compared to PN. Subsequent functional inhibition of these two micro-RNAs led to impaired Wnt/β-catenin pathways and inhibited MPNST growth. They also established a micro-RNA expression signature that could differentiate the sporadic and NF1-associated MPNSTs, indicating that micro-RNA could potentially be used as a promising diagnostic tool [67]. Zhang et al. observed an increased expression of miR-30d after the knockdown of *EZH2*, which further induced the proliferation of MPNST cells and revealed that altered micro-RNA expression could be related to tumor aggressiveness [40]. Overall, these epigenetic studies suggest that PRC2 loss and *CDKN2A* hypermethylation are common pathways that could serve as prognostic indicators or therapeutic targets for MPNSTs. Future research should leverage a combination of high-resolution methylation profiling, ATAC-seq for chromatin accessibility, and ChIP-seq for histone modifications—to map out the full landscape of epigenetic alterations in MPNSTs.

### 2.5. Emerging Sequencing Technologies: Proteomics and Metabolomic Sequencing

Recently, proteomics and metabolomics sequencing are beginning to be applied in MPNST. Tsuchiya et al. performed proteomics sequencing for 23 MPNSTs and found that the MET pathway was upregulated in the recurrent or metastatic MPNST group. Further drug screens also identified the fact that one of the MET inhibitors, crizotinib, could effectively inhibit MPNST growth, highlighting the potential to use this technology for therapeutic drug discovery [80]. Jia et al. performed proteomics sequencing of MPNST samples with different prognosis (less than 2 years’ survival vs. more than 5 years’ survival). They observed that expression of decorin, a protein maintaining the stability of extracellular matrix, was significantly lower in the poor-prognosis MPNST group, highlighting the use of this technology to identify prognostic biomarkers [81].

No large metabolomic screens have been performed. However, Lemberg et al. evaluated the effect of an inhibitor of glutamine, JHU395, on MPNST growth. JHU395 could inhibit the MPNST growth in vivo and in vitro with mild toxicity. They further analyzed the metabolomic changes following JHU395 inhibition in mice and identified formylglycinamide ribonucleotide as the most significantly altered metabolite. This metabolite plays a key role in purine synthesis, suggesting that the inhibition of purine synthesis could be the mechanism by which JHU395 impedes MPNST progression [82]. More recently, they evaluated a combination therapy of pro905 and JHU395 in MPNSTs, and detected the metabolites that change after the treatment in a mouse model. Similarly, purine and pyrimidine metabolism were still the top differentially altered pathways [83]. These studies highlight the possibility of identifying combination therapies through metabolomic analyses. Future studies should focus on cell type-specific metabolic and proteomic signatures, and within the MPNST.

## 3. From Sequencing to Implications: Targeted Therapy

### 3.1. MEK Inhibitor

MEK pathway activation has been observed in MPNSTs [52,84]. Although several studies showed promising results utilizing MEK inhibitors in benign PN, single-agent MEK inhibition has not been nearly as effective in MPNST models [32,85], likely due to adaptive signaling leading to drug resistance [86]. However, a MEK inhibitor is a potentially promising partner in combination therapies. Borcherding et al. showed the synergistic anti-tumor effect of a TYK2 inhibitor and MEK inhibitor in NF1-associated MPNST in vivo and in vitro [73]. Other promising combinations have included CDK4/6 inhibition with MEK inhibition in MPNST models. Additionally, that combination treatment increased the MPNST response rate to anti-PD-L1 immunotherapy in the mice model [87]. Wang et al. identified SHP2 as a central node in MPNST pathogenesis and the combination of an SHP2 inhibitor and MEK inhibitor looked promising in vivo, in MPNST models [88]. The above studies highlight the importance of combination therapy in MPNST to overcome drug resistance. Several clinical trials are evaluating MEK inhibitor combinations in MPNSTs (NCT05107037, NCT05253131 and NCT03433183) and other trials based on the pre-clinical studies described here are in development.

### 3.2. Histone Deacetylase (HDAC) Inhibitors

Much evidence from RNA-seq and DNA methylation sequencing has shown that PRC2 loss is a key molecular mutation event during the MPNST transformation [19,49]. PRC2 is crucial for the methylation of H3k27me3, which is the rationale for attempting to treat MPNSTs using HDAC inhibitors. Several HDAC inhibitors have been evaluated, showing promising results in vitro and in vivo in MPNST models [89,90,91]. To date, there are several HDAC inhibitors that have been approved to treat other cancers, including romidepsin, vorinostat, belinostat and panobinostat, which are currently used for lymphoma and multiple myeloma [92]. However, there is only one completed phase II clinical trial testing the effect of HDAC inhibitor romidepsin on various sarcomas (NCT00112463), and no studies to date have focused solely on MPNSTs. These findings underscore the need for further research and clinical trials to explore the potential of HDAC inhibitors in the treatment of MPNSTs, potentially in combination with other therapies.

### 3.3. Other Emerging Targeted Therapies

Several DNMT inhibitors have been evaluated in MPNSTs. Patel et al. found that a DNMT1 inhibitor selectively inhibited PRC2-loss and MPNST growth and activated transcription of a portion of PRC2 target genes [56]. Similarly, decitabine, a DNA methyltransferase inhibitor, has been approved to treat myelodysplastic syndrome [93] and an ongoing phase II clinical trial is testing the effect of ASTX727, a combination of decitabine and cedazurine, in MPNSTs (NCT04872543). These studies underscore the therapeutic potential of DNMT inhibitors in MPNST, particularly in targeting MPNSTs with PRC2 loss.

The mTOR signaling pathway has also been found to be activated in MPNSTs, and increased expression of p-mTOR is related to the poor prognosis of MPNSTs [94]. Preclinical studies have implicated mTOR signaling in MPNSTs. Johasson et al. found that the single mTOR inhibitor everolimus could inhibit the sporadic and NF1-associated MPNST growth in vivo [95]. They also showed that a combination of everolimus and erlotinib, an EGFR inhibitor, could achieve an additive effect on inhibition of tumor growth. However, clinical trial results with mTOR inhibitors have not shown benefit [96,97,98]. This may be due to poor selection of partners, specificity of mTOR inhibitors, or perhaps the fact that these combinations were studied in a single preclinical model which may not be representative of MPNSTs.

In 2013, Peacock et al. found up-regulated DNA repair mechanisms in MPNSTs, which indicates that DNA repair inhibitors might be a potential treatment for MPNSTs. Poly (ADP-ribose) polymerase (PARP) is a protein involved in DNA repair, and has been studied in MPNSTs recently. Several PARP inhibitors (e.g., olaparib, rucaparib, niraparib and talazoparib) have been approved by FDA and applied in many cancers. Larsson et al. tested a combination of trabectidin and olaparib in MPNSTs and found that this combination therapy reduced tumor growth in vivo [99]. Kivlin et al. also tested olaparib in MPNST and mouse models and found that olaparib could help decrease tumor growth and metastasis [100]. Unfortunately, no clinical trials report the effects of PARP inhibition on MPNST, and we look forward to further investigation of this target in MPNST patients.

### 3.4. Implications for Immunotherapy

Although there is currently no approved immunotherapy regimen for MPNSTs, several preclinical studies have shown that this could be a promising treatment strategy. Holand et al. identified two subtypes of MPNSTs based on transcriptome data, named immune deficient and immune active. They found that MPNSTs from an immune-deficient group had a poor prognosis and upregulated *LGR5*, *IGF2BP1*, *PROM1*, which could be used as potential druggable targets [29].

PRC2 status is also correlated with immune infiltration in MPNSTs. Yan et al. compared gene expression of MPNSTs with PRC2 retaining or loss [47]. They found that adaptive immune response pathway and T-cell receptor pathway are enriched in the PRC2-retained group while spinal cord development and other differentiation pathways are enriched in the PRC2-loss group. Further, they found that immunogenic virus injection into the PRC2-loss group could enhance the tumor immune infiltration and sensitize the tumor to immune checkpoint therapy. In addition, Cortes-Ciriano et al. identified two transcriptomic subtypes of MPNSTs, which were correlated with the H3Kme27 status, and the subtype with H3K27me3 loss had decreased immune infiltration and immune checkpoint expression inferred from RNAseq data [72].

The alteration of the immune microenvironment is being explored as a therapeutic strategy. Patwardhan et al. depleted macrophages in an MPNST xenograft model using pexidartinib, which led to reduced tumor growth [101]. In addition, Somatilaka et al. were able to cause a shift in MPNSTs from immune cold- to immune hot-tumor, using stimulator of IFN genes (STING) signaling, which sensitized the MPNST to the PD-1/PD-L1 blockade treatment [102]. Future studies could be aimed at therapeutic strategies to shift this immune cold state to an immune-rich environment, to harness immunotherapies.

## 4. Conclusions and Implications for Clinical Practice

As sequencing technologies have advanced, there has been an explosion in the volume and variety of data from MPNST, derived at multiple biological levels. These data have led to the identification of MPNST driver genes including *NF1*, *CDKN2A*, PRC2 components and *TP53*, as well as numerous potential targeted therapies including MEK inhibition combinations, HDAC inhibitors, SHP2 inhibitors, TYK2 inhibitors, and CDK4/6 inhibitors, etc. Additionally, these data have begun to uncover the potential of targeting the immune system, as well.

Given the relatively low cost of clinical sequencing platforms and potential information that can be gained to guide clinical practice, obtaining molecular information on MPNST patients is potentially beneficial. For example, *MTAP* loss, which occurs with *CDKN2A* loss in at least 25% of MPNSTs, confers sensitivity to *PRMT5* inhibitors which are currently in clinical trials [103]. Having this information would allow for patients to be enrolled in such a clinical trial. Additionally, MDM2 inhibitors are being utilized in clinical trials for MPNSTs. However, 25% of MPNSTs have *TP53* mutations, and as such, an MDM2 inhibitor would not be beneficial in this patient population [104]. Thus, molecular data can again guide decisions for clinical trial enrollment.

Despite these advancements, significant challenges remain. First, we are only starting to understand molecular subtypes of MPNSTs. For example, 25% of MPNSTs have *TP53* mutations, and approximately 65% have mutations in the PRC2 complex. However, for at least 10% of MPNSTs, the driver is unclear. Additionally, we have not had validated studies to understand whether or not there are clear transcriptomic subtypes that should guide therapy. Second, once a therapy begins, drug resistance driven by adaptive changes is bound to occur, which will certainly impact therapeutic decisions. In the coming years, development of larger cohorts in clinical trials and the integration of multi-omics data, combined with targeted therapies to overcome tumor resistance and personalized treatments, based on the molecular characteristics of MPNST, will shape the therapeutic trends of the future.

## Figures and Tables

**Figure 1 cancers-17-00180-f001:**
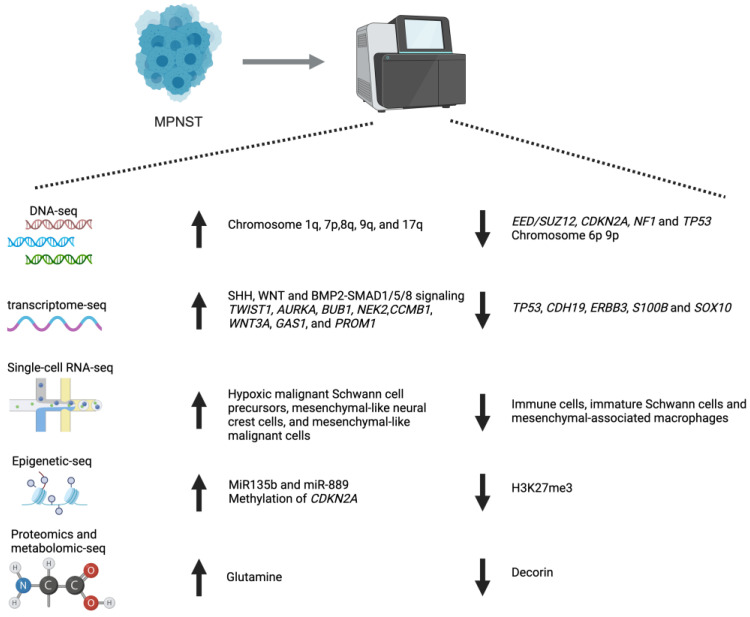
Summary of key findings from recent sequencing technologies (created by Biorender).

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
