# Peer review of "A Sequencing Overview of Malignant Peripheral Nerve Sheath Tumors: Findings and Implications for Treatment"

_cancers, 2025, doi:10.3390/cancers17020180_

Round 1
Reviewer 1 Report
Comments and Suggestions for Authors
In this review, the authors summarize the molecular analysis of MPNSTs conducted to date and the implications for clinical management and therapeutic discovery. The review is organized based on modality, beginning with microarray/RNA sequencing followed by single cell RNA sequencing, DNA sequencing, epigenetic analysis, and proteomics/metabolomics. The authors conclude by summarizing the therapeutic implications of these studies focused on MEK inhibition, HDAC inhibition and immunotherapies. Overall, this review is comprehensive and timely, particularly given the many MPNST genomic analyses published in the past 2 years.
Major Comments
1. Rather than having separate tables for each molecular assay, it would be more informative and comprehensive for the reader if this information was combined into a single table listing each study, indicating what type(s) of assays were performed, and including the relevant PMID/GSEA. Compressing this information into a single table will better facilitate comparison and re-analysis of these data.
2. In addition to the graphical abstract in Figure 1, it would be helpful to have a figure summarizing the findings from gene expression, DNA mutation, epigenetic, and single cell studies to provide the reader with a succinct snapshot of where the authors feel the field currently stands. It would also be helpful to list open questions for each modality (or additional modalities that could be applied such as metabolomics) that will be critical going forward.
Minor Comments
1. Line 45 (radiation therapy) appears to be missing a citation, and we suggest also citing PMID: 33880447 regarding the benefit of radiation for MPNSTs.
2. The “Implications for Immunotherapy” section could be expanded and could include some speculation on why MPNSTs are immune cold beyond PRC2 loss and discuss therapeutic approaches modulating the microenvironment beyond checkpoint blockade (e.g. oncolytic viruses).
Author Response
Reviewer 1: In this review, the authors summarize the molecular analysis of MPNSTs conducted to date and the implications for clinical management and therapeutic discovery. The review is organized based on modality, beginning with microarray/RNA sequencing followed by single cell RNA sequencing, DNA sequencing, epigenetic analysis, and proteomics/metabolomics. The authors conclude by summarizing the therapeutic implications of these studies focused on MEK inhibition, HDAC inhibition and immunotherapies. Overall, this review is comprehensive and timely, particularly given the many MPNST genomic analyses published in the past 2 years.
Major Comments
1) Rather than having separate tables for each molecular assay, it would be more informative and comprehensive for the reader if this information was combined into a single table listing each study, indicating what type(s) of assays were performed, and including the relevant PMID/GSEA. Compressing this information into a single table will better facilitate comparison and re-analysis of these data.
We agree that combining all tables into 1 single table will make it easier for readers to follow. We have combined it into a single table in revised manuscript (Table 1). Thank you again for your suggestions.
2) In addition to the graphical abstract in Figure 1, it would be helpful to have a figure summarizing the findings from gene expression, DNA mutation, epigenetic, and single cell studies to provide the reader with a succinct snapshot of where the authors feel the field currently stands. It would also be helpful to list open questions for each modality (or additional modalities that could be applied such as metabolomics) that will be critical going forward.
Thank you for your kind suggestions. We agree it will be better to include another figure showing the key findings throughout the MPNST progression and we have added it as figure 2 in revised manuscript.
In line 90, we have added “The summary of key findings is shown in Table 1 and Figure 2.”.
For each modality, we also list open questions that need further investigation:
In line 173, we added and highlighted “Future studies should be focused on combination with single-cell and spatial transcriptomic approaches to dissect the cellular heterogeneity within MPNST.”
In line 214, we added and highlighted “Future research should strive to improve the sensitivity of the low-abundance transcripts and simplify the pipeline of single-cell RNA-seq.”
In line 298, we added and highlighted “Future research should leverage a combination of high-resolution methylation profiling, ATAC-seq for chromatin accessibility, and ChIP-seq for histone modifications—to map out the full landscape of epigenetic alterations in MPNST.”
In line 324, we added and highlighted “Future studies should focus on cell type-specific metabolic and proteomic signatures and within MPNST.”
Minor Comments
- Line 45 (radiation therapy) appears to be missing a citation, and we suggest also citing PMID: 33880447 regarding the benefit of radiation for MPNSTs.
We apologize for this oversight. We have added the reference in the revised manuscript.
- The “Implications for Immunotherapy” section could be expanded and could include some speculation on why MPNSTs are immune cold beyond PRC2 loss and discuss therapeutic approaches modulating the microenvironment beyond checkpoint blockade (e.g. oncolytic viruses).
Thank you for your suggestions. Investigating the mechanisms underlying immune-cold MPNST is an important research focus in our lab. However, this review primarily aims to discuss sequencing technologies applied to MPNST, and while including this topic could be beneficial, it falls outside the scope of this paper. Furthermore, a colleague from our lab has already submitted a review that comprehensively addresses immunotherapy in MPNST, including the modulation of its microenvironment to enhance therapeutic efficacy. Therefore, we prefer to maintain the current focus of this paragraph. Thank you again for your suggestions!
Reviewer 2 Report
Comments and Suggestions for Authors
This review summarizes recent sequencing studies on peripheral nerve sheath 19 tumors, including plexiform neurofibromas (PN), atypical neurofibromatous neoplasm with uncertain biologic potential (ANNUBP), and MPNST.
The topic is interesting. However, the paper is extremely disorganized.
In my opinion, it would be much clearer to organize it by target. Different methods to discover mutations should be discussed in each paragraph. Thus, all tables should be re-organized accordingly. Also, would differentiate among different targets in different grade tumors.
Fig 1 unuseful
Please acknowledge the narrative nature of this review.
At the present state, although reviewing about an interesting topic, the paper is disorganized and difficult to follow.
Author Response
Reviewer 2
This review summarizes recent sequencing studies on peripheral nerve sheath 19 tumors, including plexiform neurofibromas (PN), atypical neurofibromatous neoplasm with uncertain biologic potential (ANNUBP), and MPNST.
- The topic is interesting. However, the paper is extremely disorganized.
In my opinion, it would be much clearer to organize it by target. Different methods to discover mutations should be discussed in each paragraph. Thus, all tables should be re-organized accordingly. Also, would differentiate among different targets in different grade tumors.
We are very grateful for the reviewer's suggestion that organizing it by target may result in a clearer structure. However, in this review the main goal is to summarize the recent application of sequencing technology to MPNST. After careful consideration, we decided to keep the existing review structure. We also look forward to writing a review by focusing on targets in the future as more MPNST-related targets emerge and sequencing results are further enriched.
- At the present state, although reviewing about an interesting topic, the paper is disorganized and difficult to follow.
We appreciate reviewer’s suggestion, and we added another figure summarizing the main molecular findings throughout MPNST development, which is shown as Figure 2 in the revised manuscript. We have also reorganized the review to make it easier to follow.
Reviewer 3 Report
Comments and Suggestions for Authors
In their comprehensive review, Kangwen Xiao and colleagues summarize the molecular profiling of Malignant Peripheral Nerve Sheath Tumors (MPNST). Current sequencing technologies and platforms, such as microarrays, RNA-seq, single-cell RNA sequencing, Whole Exome Sequencing (WES), and epigenetic sequencing, are discussed and effectively summarized in tables. The authors also review the implications of these technologies for targeted treatments, including MEK, HDAC, and DNMT inhibitors, as well as immunotherapies.
The review is informative, well-structured, and fits well into the special issue. There are only minor areas for improvement:
1. Line 15/35 and Others:
The assertion that all MPNSTs are “Schwann-cell derived” may oversimplify the current understanding of these tumors. It would be helpful to rephrase this statement and include a reference to ongoing pathology challenges (recent WHO classifications, or similar, like DOI:10.1093/neuonc/noae235).
2. Tables:
The tables could benefit from improved readability, but this could be addressed during the proofing stage.
3. Diagnostics and Subgrouping:
Consider discussing how new molecular profiling techniques could further refine MPNST diagnostics by identifying different cells of origin for similar looking tumors, similar to advances in medulloblastoma grouping and subgrouping (in WHO classifications 2021/2016). This could highlight the potential to define distinct molecular subtypes with different treatable targets for clinical decision making, further underscoring the clinical relevance of the review.
4. Similarity Check:
I have not checked for similarities with recent publications from this group. If required, this could be reviewed as part of the editorial process.
Author Response
Reviewer 3
In their comprehensive review, Kangwen Xiao and colleagues summarize the molecular profiling of Malignant Peripheral Nerve Sheath Tumors (MPNST). Current sequencing technologies and platforms, such as microarrays, RNA-seq, single-cell RNA sequencing, Whole Exome Sequencing (WES), and epigenetic sequencing, are discussed and effectively summarized in tables. The authors also review the implications of these technologies for targeted treatments, including MEK, HDAC, and DNMT inhibitors, as well as immunotherapies.
The review is informative, well-structured, and fits well into the special issue. There are only minor areas for improvement:
- Line 15/35 and Others:
The assertion that all MPNSTs are “Schwann-cell derived” may oversimplify the current understanding of these tumors. It would be helpful to rephrase this statement and include a reference to ongoing pathology challenges (recent WHO classifications, or similar, like DOI:10.1093/neuonc/noae235).
Thank you so much for your suggestions. We agree that further study is needed to determine the origin of the tumor. We have deleted this sentence “As a Schwann cell-derived soft tissue sarcoma,” in the abstract.
We also added and highlighted the sentence in line 37, “While MPNSTs are traditionally characterized as Schwann-cell derived tumors, this may oversimplify their heterogeneous nature. A recent study (e.g., DOI:10.1093/neuonc/noae235) highlight that low-grade MPNST should be identified as ANNUBP with increased proliferation.”
- Tables: The tables could benefit from improved readability, but this could be addressed during the proofing stage.
Thank you for your suggestions. We have reorganized the tables and combined it into a single table to make it easier for readers to follow. We also noticed the issue that the format of tables that we uploaded are not the same as the peer-review version, which might lead to the poor structure of the table. We will also contact the editor to ensure the consistency of the table formats.
- Diagnostics and Subgrouping:
Consider discussing how new molecular profiling techniques could further refine MPNST diagnostics by identifying different cells of origin for similar looking tumors, similar to advances in (successful example) medulloblastoma grouping and subgrouping (in WHO classifications 2021/2016). This could highlight the potential to define distinct molecular subtypes with different treatable targets for clinical decision making, further underscoring the clinical relevance of the review.
Thank you for your constructive suggestions. We have added and highlighted it in line 113: “Molecular classification has become increasingly important in tumor diagnostics and treatment. For instance, in 2016, medulloblastoma was subdivided into multiple mo-lecular subtypes, including the Sonic Hedgehog (SHH) and WNT subtypes, based on transcriptomic data[70]. This subtype-specific classification has since enabled the development of tailored therapeutic strategies, significantly improving patient prognoses[71].”
Round 2
Reviewer 2 Report
Comments and Suggestions for Authors
I regret the Authors were not able to address appropriately to any of my previous concerns. Thus, I still believe that the paper is not suitable for the publication.
Very disorganized, difficult to follow.
Comments on the Quality of English LanguageStill many grammar and syntax errors.
Author Response
Thank you for taking the time to review our manuscript again. We appreciate your honest feedback and understand your concerns. While we regret that our revisions did not meet your expectations, we value your input you have taken in evaluating our work. Thank you once more for your time.